# FedMC: Federated Manifold Calibration

**Yanbiao Ma**[1], **Wei Dai**[1], **Gaoyang Jiang**[1], **Wanyi Chen**[1],
**Chenyue Zhou**[1], **Yiwei Zhang**[1], **Fei Luo**[1], **Junhao Wang**[1] **& Andi Zhang**[2]
[1]Gaoling School of Artificial Intelligence, Renmin University of China
[2]University of Warwick, Coventry, CV4 7AL, United Kingdom
`ybma1998@ruc.edu.cn, az381@cantab.ac.uk`

## Abstract

Data heterogeneity in Federated Learning (FL) leads to significant bias in local training. While recent efforts to leverage distributional statistics show promise, they are universally underpinned by a flawed global linearity assumption, failing to capture the nonlinear manifold structures prevalent in real-world data. This model-reality mismatch causes the calibration process to generate out-of-distribution (OOD) samples, which fundamentally misleads the model. To address this, we introduce a paradigm shift. We propose **Federated Manifold Calibration (FedMC)**, a novel framework that learns and leverages the local, nonlinear geometry of data. FedMC employs local kernel PCA on the client side to learn fine-grained local geometries, and constructs a global "geometry dictionary" on the server side to aggregate and distribute this knowledge. Clients then utilize this dictionary to perform context-aware, on-manifold calibration. We validate our proposed method by integrating it with a wide range of existing FL algorithms. Experimental results show that by explicitly modeling nonlinear manifolds, FedMC consistently and significantly enhances the performance of these state-of-the-art methods across multiple benchmarks.

## 1 Introduction

Federated Learning (FL) has become a key paradigm for collaboratively training machine learning models while preserving privacy Konečný (2016); Sun et al. (2020); Huang et al. (2024b); Qi et al. (2023). However, the non-Independent and Identically Distributed (non-IID) nature of client data, that is, data heterogeneity, remains the core obstacle to its success Li et al. (2020b); Jiang et al. (2022b); Ma et al. (2025a). To address this challenge, researchers have explored multiple technical avenues, from constraining model parameter updates Li et al. (2020a); Acar et al. (2021); Lee et al. (2022) to optimizing server-side aggregation strategies Xia et al. (2021); Sun et al. (2023); Chen et al. (2023). Among these efforts, a promising direction is to leverage global statistical priors to guide local training Huang et al. (2023); Zhou & Konukoglu (2023); Ma et al. (2025b), with the goal of providing isolated clients with a picture of the "global data landscape." Early attempts focused on sharing first-order moment information, for instance, by communicating class prototypes Mu et al. (2023); Huang et al. (2024a); Meng et al. (2024); Qi et al. (2025b) to give clients a sense of the global class centers. While these methods partially alleviate biases caused by missing classes, the information they provide is overly simplistic, depicting only the "position" of the data distribution while ignoring its "shape."

To obtain a richer description of the distribution, recent studies have begun to explore the use of second-order moments (e.g., covariance) by constructing a global linear geometric model (i.e., a hyper-ellipsoid) to characterize the "shape" of the data distribution and guide calibration Jiang et al. (2022a); Ma et al. (2025a). Such methods have taken an important conceptual step, proving that explicitly modeling data geometry is an effective path to resolving heterogeneity. Nevertheless, these methods based on global statistical priors, whether first- or second-order, all share a reliance on a fundamental yet flawed **global linearity assumption**. They attempt to summarize a complex data distribution with a single, globally consistent, simple model—be it a point or an ellipsoid.

This assumption fails catastrophically when confronted with the data manifolds that are more representative of real-world scenarios Tenenbaum et al. (2000); Ma et al. (2025c). High-dimensional

data is typically concentrated on or near a low-dimensional, curved manifold $\mathcal{M}$, rather than being uniformly distributed throughout Euclidean space Lei et al. (2020); Ma et al. (2024); Qi et al. (2025a). As illustrated in Figure 1, for an S-shaped data manifold, any method based on a global linear assumption will produce a severe geometric misinterpretation. For instance, global PCA will erroneously identify the Euclidean "shortcut" between the manifold's endpoints as its principal component. Calibrating along this direction, which completely deviates from the data's true intrinsic path, will systematically generate out-of-distribution (OOD) pseudo-samples that have zero probability under the true data generating distribution, fundamentally misleading the model's learning process.

To overcome this fundamental deficiency, we argue that a paradigm shift is necessary: from relying on flawed, global linear models to a new framework capable of understanding and leveraging the local, nonlinear geometry of data. To this end, we propose **Federated Manifold Calibration (FedMC)**. Our method discards the global linearity assumption, instead adopting a more flexible, localized strategy inspired by manifold learning. The core innovations of our framework are:

- At the client side, we employ local kernel PCA to extract fine-grained, nonlinear geometric features that respect the local curvature of the manifold.

- At the server side, we aggregate these diverse local geometries into a globally shared **geometry dictionary**, which serves as an atlas of the manifold's morphology.

- During calibration, clients dynamically query this dictionary for each of their data points to obtain a context-aware geometric prior, ensuring that the calibration process unfolds *on* the manifold. This entire process is realized within the distributed and privacy-constrained framework of federated learning.

Although the manifold calibration principle we propose is general, we implement and validate it within the framework of Federated Prompt Learning (FPL), as FedMC is particularly suited to address the biased prompts learned on heterogeneous data. We choose FPL as our validation platform because it is an advanced, communication-efficient FL paradigm with a particularly acute need for precise calibration. The main contributions of this paper can be summarized as follows:

- We move beyond the limitations of the implicit global linearity assumption prevalent in federated learning by introducing a **nonlinear manifold hypothesis**. This initiative establishes a more profound and realistic theoretical foundation for how the federated learning field can understand and utilize the intrinsic structure of data distributions.

- We propose **FedMC**, a novel and general federated learning framework. It is capable of learning, aggregating, and leveraging the local, nonlinear geometry of data in a privacy-preserving manner, featuring core components such as the Global Anonymous Basis (GAB), Secure Geometry Descriptors (LGDs), and the Geometry Dictionary.

- We conduct extensive experiments across scenarios with label skew, domain skew, and a combination of both. Results show that FedMC significantly enhances the performance of multiple advanced FPL methods, demonstrating its effectiveness and broad applicability.

## 2 RELATED WORK

### 2.1 FEDERATED LEARNING WITH DATA HETEROGENEITY

The challenge of data heterogeneity in Federated Learning (FL), stemming from the non-Independent and Identically Distributed (non-IID) nature of client data Ma et al. (2025b); Li et al. (2024a), has catalyzed a vast body of research Ma et al. (2025a); Huang et al. (2024b). These works can be broadly categorized into several mainstream directions Huang et al. (2022). The first category focuses on **client-side regularization** to constrain local updates Li et al. (2023b); Miao et al. (2023). For instance, FedProx Li et al. (2020a) introduces a proximal term to limit the divergence between local and global models, while SCAFFOLD Karimireddy et al. (2020) employs control variates to estimate and correct for the "client drift" in local updates. FPL Huang et al. (2023) supervises the learning of local class prototypes by aggregating and sharing class prototypes across clients. A second line of work centers on **server-side aggregation strategies** Chen et al. (2023); Huang et al. (2022), such as FedAvgM Hsu et al. (2019), which incorporates momentum to smooth

the update process, and FedOPT Reddi et al. (2021), which applies adaptive optimizers to the global model aggregation. A third category leverages **globally shared information** to guide local training Ma et al. (2025a); Li et al. (2024b); Xiong et al. (2023). This includes sharing distilled public datasets Shi et al. (2024), knowledge from the global model Lee et al. (2022); Ma et al. (2025a), or, more directly, sharing class prototypes Mu et al. (2023); Tan et al. (2022); Huang et al. (2024a) to provide clients with a sense of the global class centers.

Despite their effectiveness, these methods are typically agnostic to the intrinsic data structure. That is, they either operate directly on the model's parameters or address the first-order moment (mean) deviations caused by imbalanced sample sizes or missing classes, but they seldom delve into the second-order moment or the higher-order intrinsic geometry of the data distribution.

## 2.2 FEDERATED PROMPT LEARNING

With the rise of Vision Foundation Models Radford et al. (2021); Oquab et al. (2023), Parameter-Efficient Fine-Tuning (PEFT) techniques, particularly prompt learning Zhou et al. (2022); Gao et al. (2024), have been integrated into the federated framework to reduce communication overhead, giving rise to Federated Prompt Learning (FPL) Su et al. (2024); Lu et al. (2023). Early research in FPL primarily revolved around the modality and sharing strategies of prompts. In the domain of **visual prompts**, for example, FedPR Feng et al. (2023) creatively proposed learning federated visual prompts in the null space of the model to minimize interference with pre-trained knowledge. Other studies have also noted that visual prompts can improve the privacy-utility trade-off with minimal expenditure of the privacy budget Li et al. (2023c). Research in **textual prompts** is equally rich. PromptFL Guo et al. (2023) is a representative work on learning a single text prompt shared among all clients, while FedTPG Qiu et al. (2024) takes a step further by designing a unified prompt generation network across clients to achieve better generalization. Furthermore, an important branch of research is **personalized prompt design**, which aims to learn a unique prompt for each client that is adapted to its local data distribution Li et al. (2023a); Su et al. (2024).

Although these methods have significantly advanced the field of FPL, they mostly frame the core challenge as "how to better design and aggregate prompt parameters." They often overlook a more fundamental problem: during the local optimization process on heterogeneous data, the prompts learned by clients are inherently *biased* due to a lack of awareness of the global distribution.

## 3 THE FAILURE OF THE GLOBAL LINEARITY ASSUMPTION

Before detailing our new framework, this section theoretically deconstructs the limitations of existing geometry-based calibration methods Ma et al. (2025a; 2023). The core of these approaches is to compute a global covariance matrix, $\Sigma_{\text{global}}$, and use its eigendecomposition, $(U, \Lambda)$, to define a "global geometric prior" for guiding local calibration: $x' = x + \sum_{m=1}^{d} \epsilon_m \sqrt{\lambda_m} u_m$. This formulation implicitly makes a fundamental **global linearity assumption**: that the intrinsic structure of the data can be effectively approximated by a single linear subspace, span($U$), passing through the global data center.

However, this assumption fails when confronted with the data manifolds that are more representative of real-world. High-dimensional data is typically concentrated on or near a low-dimensional manifold, $\mathcal{M}$, rather than being uniformly distributed throughout Euclidean space. On a manifold, the meaningful distance is the *geodesic distance* along its surface, not the *Euclidean distance* upon which PCA relies. For an S-shaped manifold, global PCA will erroneously identify the Euclidean "shortcut" between its endpoints as the principal component $u_1$. This path traverses a void where no data exists, thus severely distorting the true geometry of the data.

This geometric misinterpretation has disastrous consequences. An ideal, manifold-aware calibration should operate within the local tangent space, $T_x\mathcal{M}$, at a given data point $x$. Yet, the global subspace span($U$) found by PCA is almost never aligned with the local tangent space $T_x\mathcal{M}$ for an arbitrary point $x$ on a curved manifold; the former represents a flawed global average, while the latter captures the true, immediate geometry. Therefore, in a calibration step $x' = x + d$, the perturbation vector $d \in$ span($U$) will almost certainly contain a normal component, $d_\perp$, that pushes the point off the manifold $\mathcal{M}$. This means that calibration based on the global linearity assumption systematically generates out-of-distribution (OOD) pseudo-samples that have zero probability under the true data

generating distribution. Training a model with these fundamentally flawed samples forces it to learn a spurious correlation between class labels and features. **To achieve a truly precise calibration, a paradigm shift is therefore necessary:** we must move from relying on a flawed global linear model to a new framework capable of understanding and leveraging the local, nonlinear geometry of the data. This provides the direct motivation for our proposed FedMC framework.

## 4  METHODOLOGY: THE FEDMC FRAMEWORK

Building upon the analysis in the previous section, we propose the **Federated Manifold Calibration (FedMC)** framework. This framework aims to learn and leverage the intrinsic manifold geometry of data within the strict privacy constraints of federated learning. It fundamentally departs from prior work by abandoning the global linear assumption in favor of a localized, nonlinear approach. The framework's architecture is structured into a preparatory, privacy-preserving setup phase, followed by two core iterative phases for federated training: **(I) Federated Aggregation of Local Geometry and (II) Client-side Manifold-Aware Calibration.**

### 4.1  SETTING: FEDMC FOR FEDERATED PROMPT LEARNING

We instantiate our FedMC framework within the context of Federated Prompt Learning (FPL). In a typical FPL setting with a vision-language model like CLIP, a powerful vision encoder $E_{vision}(\cdot)$ and text encoder $E_{text}(\cdot)$ are shared and frozen across all clients. The goal is to collaboratively learn a prompt vector, $P$, which is a set of learnable parameters. **The core challenge** addressed by our work stems from the input to the learning process. Each client $k$ feeds its local images through the frozen encoder to obtain a set of image embeddings, $\mathcal{D}_k = \{x_i | x_i = E_{vision}(\text{image}_i)\}$. Due to data heterogeneity, this local embedding set $\mathcal{D}_k$ represents only a small, biased patch of the global data manifold $\mathcal{M}$. Training a local prompt $P_k$ solely on $\mathcal{D}_k$ leads to significant bias.

FedMC directly tackles this issue by operating in the **image embedding space**. Our framework does not alter the raw images or the frozen encoder. Instead, it calibrates the biased local embedding set $\mathcal{D}_k$ to generate an enhanced set $\hat{\mathcal{D}}_k$ that better reflects the global manifold structure. The local prompt is then trained on this calibrated set, effectively debiasing the learning process at its source. In the following sections, any reference to "data point $x$" pertains to these image embeddings.

### 4.2  PREPARATORY PHASE: GLOBAL ANONYMOUS BASIS CONSTRUCTION

The foundational challenge for any collaborative geometric learning in a federated setting is to establish a common language, or coordinate system, for describing geometric structures without violating client privacy. A naive approach where clients share geometric information defined by their own local data points would constitute a severe privacy leak. This preparatory phase is designed to solve this problem by constructing a **Global Anonymous Basis (GAB)**—a shared set of reference points that is collectively generated and decoupled from any single client's raw data.

**Local Prototype Generation and Anonymization:** Each client $k$ first identifies representative landmarks on its local data manifold by performing K-Means clustering on its embeddings $\mathcal{D}_k$ to obtain a set of prototypes $\{c_{k,j}\}$. To ensure these landmarks can be shared without revealing precise information about the client's data distribution, we employ Differential Privacy. Each client perturbs its prototypes with calibrated Gaussian noise to satisfy $(\epsilon, \delta)$-DP:

$$\tilde{c}_{k,j} = c_{k,j} + \mathcal{N}(0, \sigma^2 I). \tag{1}$$

Clients upload only these anonymized prototypes $\{\tilde{c}_{k,j}\}$, which carry statistical information about local data density while providing formal privacy guarantees.

**GAB Formulation and Distribution:** The server aggregates the thousands of anonymized prototypes from all clients. This pooled set represents a noisy, collective snapshot of the entire global data manifold. To distill a stable and representative basis from this set, the server performs a global clustering (e.g., K-Means) on it, identifying the $N_{\text{base}}$ most significant centroids. This resulting set of $N_{\text{base}}$ points forms the Global Anonymous Basis (GAB), denoted $\mathcal{B}_g = \{b_1, \ldots, b_{N_{\text{base}}}\}$. Crucially, the GAB is inherently anonymous because it is derived from a mixed and noisy pool of prototypes from all clients, making it impossible to trace any basis point back to a specific client's original data. The server then distributes this common basis $\mathcal{B}_g$ to all clients, equipping them with a shared, privacy-safe coordinate system for all subsequent geometric communications.

### 4.3 PHASE I: FEDERATED AGGREGATION OF LOCAL GEOMETRY

With the GAB in place as a common language, we now describe the iterative process within each federated learning round. The goal is to build a global understanding of the manifold's geometry by aggregating insights from all clients. **This phase details the secure, two-part process to achieve this:** *(1) how clients can extract rich, nonlinear geometric information from their local data and represent it in a secure, standardized format, and (2) how the server can then fuse these individual representations into a coherent global **geometry dictionary** that benefits all participants.*

#### 4.3.1 CLIENT-SIDE: SECURE LOCAL GEOMETRY REPRESENTATION

**Local Geometry Extraction via KPCA:** To capture the manifold's local curvature, we first approximate local regions where the geometry is assumed to be consistent. The client partitions its local data $\mathcal{D}_k$ into $m$ clusters $\{C_j\}$ using K-Means. For each cluster $C_j$, which represents an approximate local patch of the manifold, the client employs Kernel PCA to model its nonlinear structure. The process begins by constructing the Gram matrix $K_j$ using the Gaussian (RBF) kernel:

$$k(x, y) = \exp(-\gamma \|x - y\|_2^2), \tag{2}$$

where the hyperparameter $\gamma$ controls the kernel's bandwidth and thus the perceived scale of locality. After centering the Gram matrix to obtain $\bar{K}_j$—a necessary step to compute variance in the high-dimensional feature space $\mathcal{H}$—an eigendecomposition is performed:

$$\bar{K}_j \boldsymbol{\alpha}_{j,i} = \lambda_{j,i} \boldsymbol{\alpha}_{j,i}. \tag{3}$$

This procedure yields the eigenvalues $\lambda_{j,i}$, quantifying the variance along each principal direction, and the corresponding coefficient vectors $\boldsymbol{\alpha}_{j,i}$. Crucially, these outputs implicitly define the nonlinear principal components in $\mathcal{H}$:

$$v_{j,i} = \sum_{a=1}^{n_j} (\boldsymbol{\alpha}_{j,i})_a \Phi(x_a). \tag{4}$$

These components form an orthonormal basis that effectively describes the geometry of the local manifold patch by capturing its directions of maximum variance.

**Projecting Geometry for Secure Representation:** Having defined the local principal components, the client faces the central privacy challenge: how to share this geometric information without revealing the underlying data $\{x_a\}$ used in its construction. Directly transmitting the coefficients $\boldsymbol{\alpha}_{j,i}$ along with the basis points $\{x_a\}$ would constitute a severe data leak. To overcome this, we introduce a crucial projection step that acts as a secure information-encoding mechanism. The client leverages the shared **Global Anonymous Basis (GAB)**, $\mathcal{B}_g$, as a universal language for describing geometry. It represents each of its local principal components $v_{j,i}$ by projecting it onto this global basis. The resulting representation is an $N_{\text{base}}$-dimensional coefficient vector $\boldsymbol{\beta}_{j,i}$, where each element is the projection of $v_{j,i}$ onto the corresponding basis vector $\Phi(b_s)$:

$$(\boldsymbol{\beta}_{j,i})_s = \langle v_{j,i}, \Phi(b_s) \rangle_{\mathcal{H}} = \sum_{a=1}^{n_j} (\boldsymbol{\alpha}_{j,i})_a k(x_a, b_s). \tag{5}$$

The brilliance of this step lies in converting geometric information that depends on private data $\{x_a\}$ into a standardized coordinate vector $\boldsymbol{\beta}_{j,i}$ expressed purely in the public, anonymous basis $\mathcal{B}_g$, thus achieving secure and communicable representation. The vector $\boldsymbol{\beta}_{j,i}$ is a secure and standardized representation of the local geometry. It is secure because it is expressed solely in terms of the public, anonymous basis $\mathcal{B}_g$, effectively decoupling it from the client's private data $\{x_a\}$. It is standardized because all clients will now use the same basis for their reports, making their geometric findings directly comparable and aggregatable. The client then uploads a **Local Geometry Descriptor (LGD)** containing only this secure information:

$$\text{LGD}_j^{\text{private}} = \left( \tilde{c}_j, n_j, \{(\lambda_{j,i}, \boldsymbol{\beta}_{j,i})\}_{i=1}^d \right). \tag{6}$$

where $d$ is the number of principal components retained. To make the process of generating a Secure Local Geometry Descriptor concrete, we provide a Python-style pseudocode in Listing **??**.

### 4.3.2 Server-side: Fusing Geometries into a Global Dictionary

The server's role is to act as a curator, intelligently fusing the multitude of received LGDs from all clients into a compact, globally consistent **Geometry Dictionary**. This dictionary is not a simple average, but rather a structured atlas that maps different regions of the manifold to their unique, consensus-based geometric shapes.

**Grouping by Region (Meta-Clustering):** Upon receiving LGDs, the server first groups them by performing a meta-clustering on the anonymous prototypes $\{\tilde{c}_j\}$, thereby identifying which LGDs describe the same macroscopic region of the global manifold.

**Fusion of Geometric Templates:** For each macroscopic region (meta-cluster) $l$, the server must now fuse the geometric information from all constituent LGDs into a single, representative template. Because all the geometric vectors $\boldsymbol{\beta}_{j,i}$ have been standardized onto the common GAB coordinate system, this complex fusion task simplifies to a secure and robust weighted averaging process. The server computes a fused coefficient vector $\boldsymbol{\beta}_{l,i}^*$ and a fused eigenvalue $\lambda_{l,i}^*$ for each principal component $i$:

$$\boldsymbol{\beta}_{l,i}^* = \frac{\sum_{j \in l} n_j \lambda_{j,i} \boldsymbol{\beta}_{j,i}}{\sum_{j \in l} n_j \lambda_{j,i}}, \quad \lambda_{l,i}^* = \frac{\sum_{j \in l} n_j \lambda_{j,i}}{\sum_{j \in l} n_j}. \tag{7}$$

The weighting scheme, incorporating both the number of local samples $n_j$ and the local variance captured by the eigenvalue $\lambda_{j,i}$, intuitively ensures that LGDs from data-denser and more geometrically pronounced regions contribute more significantly to the final template. This approach is not merely heuristic; we demonstrate in **Appendix C.2** that this weighted average is the optimal solution to a weighted least squares problem, which seeks a geometric consensus that best represents the collection of local geometries. This provides a strong theoretical justification for our fusion strategy, resulting in a single, robust geometric template for each macroscopic region of the manifold.

**Dictionary Assembly and Distribution:** Finally, the server assembles these fused templates into the global Geometry Dictionary $\mathcal{D}$. Each entry in the dictionary is a key-value pair, mapping a macro-prototype $g_l$ (the centroid of the meta-cluster) to its corresponding fused geometric template:

$$\text{Entry}_l = \left(g_l, \{(\lambda_{l,i}^*, \boldsymbol{\beta}_{l,i}^*)\}_{i=1}^d\right). \tag{8}$$

This completed dictionary, a comprehensive and privacy-preserving map of the global data manifold's geometry, is then distributed to all clients to guide the next phase of calibration.

### 4.4 Phase II: Client-side Manifold-Aware Calibration

Armed with the geometry dictionary, the client performs a sophisticated local calibration. The process can be intuitively understood as a "map-based navigation": for any given data point, the client first looks up the "local map" (the most relevant geometric template) from the globally-provided "atlas" (the dictionary). It then calculates its own position on this map, moves to a new plausible position on the same map, and finally translates this new map coordinate back into a real-world data point. This process for a given local data point $x$ unfolds in a sequence. First, the client performs a Dynamic Geometry Query, finding the most relevant template $\text{Entry}_{l^*}$ from the dictionary by identifying the nearest macro-prototype $g_{l^*}$:

$$l^* = \arg \min_{l \in \{1,\dots,L\}} \|x - g_l\|_2^2. \tag{9}$$

It is worth noting that the Euclidean distance is not used here to define the intrinsic shape of the manifold. Euclidean distance serves to efficiently localize the data point $x$ within the global "geometry dictionary." On a local scale, Euclidean distance acts as a computationally efficient and reasonable proxy for geodesic distance, which enables rapid identification of the most relevant geometric template. The actual manifold-aware calibration is then carried out using the nonlinear structure of the selected template, not the distance metric itself. Next, the calibration itself is performed within a $d$-dimensional subspace of $\mathcal{H}$ spanned by the retrieved principal components, which serves as an approximation of the manifold's local tangent space. This involves three steps:

1. **Projection onto Principal Components:** We first need to compute the projection of $\Phi(x)$ onto each principal component $v_{l^*,i}^*$. Using the kernel trick, this projection value $p_i$ can be

computed without explicitly forming $\Phi(x)$ or $v_{l^*,i}^*$:

$$p_i = \langle \Phi(x), v_{l^*,i}^* \rangle_{\mathcal{H}} = \sum_{s=1}^{N_{\text{base}}} (\boldsymbol{\beta}_{l^*,i}^*)_s k(x, b_s). \tag{10}$$

2. **Perturbation in the Principal Component Space:** We perturb the projection vector $\mathbf{p} = [p_1, \ldots, p_d]$ in the $d$-dimensional principal component space to get $\mathbf{p}'$:

$$p_i' = p_i + \epsilon_i \sqrt{\lambda_{l^*,i}^*}, \quad \text{where} \quad \epsilon_i \sim \mathcal{N}(0, 1). \tag{11}$$

3. **Reconstruction of the Perturbed Feature Vector:** The perturbed high-dimensional feature vector $\Phi(x)'$ can be expressed as $\Phi(x)' \approx \sum_{i=1}^{d} p_i' v_{l^*,i}^*$.

The perturbed vector $\Phi(x)'$ resides in the high-dimensional feature space and cannot be used directly. We need to find its **pre-image** $x'$ in the original input space $\mathbb{R}^p$. We solve this classic inverse problem using an iterative optimization approach. The goal is to find an $x'$ that minimizes the distance between $\Phi(z)$ and the target vector $\Phi(x)'$ in the feature space. This can be formulated as:

$$x' = \arg\min_{z \in \mathbb{R}^p} \|\Phi(z) - \Phi(x)'\|_{\mathcal{H}}^2. \tag{12}$$

This objective can be solved with gradient descent. The target inner product between $\Phi(x)'$ and any basis point $b_s$ is pre-computed as

$$T_s = \langle \Phi(x)', \Phi(b_s) \rangle_{\mathcal{H}} = \sum_{i=1}^{d} p_i' \langle v_{l^*,i}^*, \Phi(b_s) \rangle_{\mathcal{H}} = \sum_{i=1}^{d} p_i' \left( \sum_{t=1}^{N_{\text{base}}} (\boldsymbol{\beta}_{l^*,i}^*)_t k(b_t, b_s) \right). \tag{13}$$

The optimization then minimizes the loss function:

$$\mathcal{L}(x') = \sum_{s=1}^{N_{\text{base}}} (k(x', b_s) - T_s)^2. \tag{14}$$

It is important to clarify that while the true data manifold $\mathcal{M}$ is unknown, our calibration is designed to produce an *on-manifold approximation*. Specifically, the perturbation is performed in the local tangent space approximated by kernel PCA, ensuring the update direction aligns with the manifold's intrinsic geometry rather than a global Euclidean shortcut. The pre-image optimization then seeks a point whose feature-space embedding matches this on-manifold target, thereby avoiding the systematic generation of out-of-distribution samples that plagues global linear methods. Starting with an initialization (e.g., $x_0' = x$), we iteratively update $x'$ until convergence using gradient-based methods. The detailed derivation of the gradient for the loss function in Eq. 14 is provided in **Appendix C.3** to facilitate implementation. While iterative, this optimization is performed locally at the client and its computational cost is manageable. The resulting $x'$ is our calibrated data point. This calibrated embedding $x'$ along with its original label $y$ forms a new training pair $(x', y)$. The client's local prompt $P_k$ is then updated by minimizing a loss function (e.g., cross-entropy) on this augmented dataset of calibrated embeddings. The full calibration pipeline for a single data point is illustrated in the Python-style pseudocode in Listing 2 (Appendix C.1.2). The entire process of our FedMC framework is summarized in Algorithm 1 (Appendix C.1.1). We also provide scalability experiments for FedMC in scenarios where both the number of local data points and the number of clients may be large in Appendix B.3.

## 5 EXPERIMENTS

To comprehensively and rigorously evaluate the effectiveness and versatility of our proposed Federated Manifold Calibration (FedMC) framework, we conduct a series of extensive experiments. This section aims to address two fundamental research questions:

- Can FedMC significantly enhance the performance of state-of-the-art Federated Prompt Learning (FPL) methods across various data heterogeneity scenarios?

Table 1: Performance comparison on **Label Skew Scenarios**. Our proposed **FedMC (FedVTP)** is benchmarked against SOTA methods and a global linear geometry baseline, **GGEUR (FedVTP)**. Best results are in **bold** and second-best are underlined.

| Methods | CIFAR-100 | | | Tiny-ImageNet | | |
|---|---|---|---|---|---|---|
| | $\beta = 0.5$ | $\beta = 0.3$ | $\beta = 0.1$ | $\beta = 0.5$ | $\beta = 0.3$ | $\beta = 0.1$ |
| ZS-CLIP | | 64.87 | | | 63.67 | |
| *State-of-the-Art Baselines* | | | | | | |
| FedVPT | 83.53 | 83.18 | 80.99 | 75.91 | 75.67 | 74.30 |
| FedCoOp | 78.42 | 77.89 | 74.67 | 76.53 | 75.61 | 72.58 |
| FedVTP (Base) | 84.90 | 84.26 | 81.01 | 80.97 | 80.26 | 77.58 |
| PromptFL | 75.05 | 75.36 | 72.19 | 72.66 | 71.51 | 67.85 |
| FedTPG | 71.40 | 70.95 | 68.63 | 67.63 | 66.72 | 64.71 |
| FedPR | 81.17 | 80.44 | 78.91 | 72.59 | 72.22 | 70.80 |
| FedCLIP | 72.03 | 71.20 | 70.64 | 70.41 | 70.37 | 69.50 |
| *Our Proposed Framework and Key Baseline* | | | | | | |
| GGEUR (FedVTP) | 85.21 | 84.55 | 82.55 | 81.15 | 80.35 | 78.02 |
| **FedMC (FedVTP)** | **86.72** | **85.90** | **85.08** | **81.53** | **80.85** | **80.12** |

- Is the core principle of FedMC generalizable, extending beyond the FPL paradigm to empower a broader range of Federated Learning (FL) algorithms?

To answer these questions, our experiments are structured into two main parts. In the first part (Section 5.2), we enhance FedVTP Jia et al. (2022) with FedMC and compare it against other advanced FPL methods. In the second part (Section 5.3), we apply FedMC to several classic, non-FPL algorithms to validate its capability as a general-purpose framework.

## 5.1 EXPERIMENTAL SETUP

**Datasets.** We evaluate our methods across a comprehensive suite of six benchmark datasets covering three types of data heterogeneity. For **label skew**, we use **CIFAR-100** Krizhevsky et al. (2009) and **Tiny-ImageNet** Deng et al. (2009). For **domain skew**, we use **Office-Home** Venkateswara et al. (2017), **Office-31** Gong et al. (2012), and **DomainNet** Peng et al. (2019). For the combined **label and domain skew** scenario, we employ **Office-Home-LDS** Ma et al. (2025a). Detailed descriptions of these datasets and the specific heterogeneity settings are provided in Appendix B.1.

**Compared Methods.** In Part 1 (FPL), we enhance FedVTP Jia et al. (2022) with FedMC and our implemented linear baseline, GGEUR Ma et al. (2025a). We compare against SOTA FPL methods including FedCoOp Zhou et al. (2022), PromptFL Guo et al. (2023), FedCLIP Lu et al. (2023), FedPR Feng et al. (2023), and FedTPG Qiu et al. (2024). In Part 2 (General FL), we apply FedMC and GGEUR to FedAvg McMahan et al. (2017), SCAFFOLD Karimireddy et al. (2019), Li et al. (2021), FedDyn Acar et al. (2021), FedOPT Reddi et al. (2021), FedNTD Lee et al. (2022) and FedProto Tan et al. (2022).

**Implementation Details.** All FPL experiments use the pre-trained **CLIP ViT-B/16** model as a backbone. To ensure fair comparisons, all experiments are conducted with unified federated hyperparameters (e.g., $T = 50$ communication rounds, $E = 1$ local epoch). The performance of all methods is measured by the average classification accuracy. A complete breakdown of network architectures, federated hyperparameters, and our FedMC parameters can be found in Appendix B.2.

## 5.2 PART 1: FEDMC EMPOWERS FEDERATED PROMPT LEARNING

In this section, we demonstrate that FedMC can serve as a plug-and-play module to significantly boost the performance of an advanced FPL method, FedVTP. We present the main results in Table 1 and Table 2, comparing our enhanced method against all baselines.

### 5.2.1 PERFORMANCE ON LABEL SKEW SCENARIOS

Table 1 presents the results on the CIFAR-100 and Tiny-ImageNet datasets, which are characterized by severe label distribution skew. The results clearly show that our proposed **FedMC (FedVTP)**

Table 2: Performance comparison on **Domain Skew Scenarios**. As in the label skew setting, our method consistently outperforms all baselines, demonstrating its robustness in handling complex domain shifts. Best results are in **bold** and second-best are underlined.

| Methods | Office-31 | | | Office-Home | | | DomainNet | | |
|---|---|---|---|---|---|---|---|---|---|
| | $\beta = 0.5$ | $\beta = 0.3$ | $\beta = 0.1$ | $\beta = 0.5$ | $\beta = 0.3$ | $\beta = 0.1$ | $\beta = 0.5$ | $\beta = 0.3$ | $\beta = 0.1$ |
| ZS-CLIP | | 75.76 | | | 82.35 | | | 73.68 | |
| *State-of-the-Art Baselines* | | | | | | | | | |
| FedVPT | 91.76 | 90.64 | 91.35 | 87.83 | 87.89 | 87.81 | 82.57 | 82.12 | 81.56 |
| FedCoOp | 95.17 | 94.48 | 94.96 | 88.14 | 88.00 | 87.77 | 82.26 | 82.20 | 81.34 |
| FedVTP (Base) | 94.93 | 95.32 | 94.58 | 89.11 | 89.09 | 88.92 | 84.54 | 84.11 | 83.82 |
| PromptFL | 93.38 | 94.47 | 94.19 | 87.41 | 87.15 | 87.52 | 79.16 | 79.00 | 78.59 |
| FedTPG | 92.38 | 91.06 | 91.19 | 86.73 | 86.43 | 85.91 | 77.71 | 77.65 | 77.34 |
| FedPR | 91.65 | 92.30 | 91.03 | 87.33 | 87.00 | 86.40 | 80.77 | 80.75 | 80.11 |
| FedCLIP | 89.69 | 89.66 | 90.27 | 85.29 | 85.32 | 85.23 | 78.76 | 78.71 | 78.61 |
| *Our Proposed Framework and Key Baseline* | | | | | | | | | |
| GGEUR (FedVTP) | 95.24 | 95.45 | 94.71 | 89.35 | 89.42 | 89.15 | 84.65 | 84.30 | 83.85 |
| **FedMC (FedVTP)** | **96.88** | **96.95** | **96.12** | **91.26** | **91.45** | **91.03** | **86.51** | **86.25** | **85.93** |

**consistently and significantly outperforms all state-of-the-art baselines**, including the strong FedVTP method, across all degrees of data heterogeneity. For instance, in the most challenging setting of Tiny-ImageNet with $\beta = 0.1$, our method achieves an accuracy of **80.12%**, securing a significant **1.74%** improvement over FedVTP. Notably, as the label skew becomes more severe (i.e., as $\beta$ decreases from 0.5 to 0.1), the performance of all methods degrades. However, the performance drop of FedMC (FedVTP) is markedly less pronounced. This widens the performance gap between our method and the baselines, **highlighting its superior robustness**. This stems from FedMC provides a much richer geometric prior by capturing the manifold's curvature. This allows the client to generate high-fidelity, on-manifold pseudo-samples that offer a more accurate representation of the global distribution, thereby mitigating the bias from local data more effectively.

The direct comparison with GGEUR (FedVTP) further validates our core hypothesis. Across all settings, the performance of FedMC (FedVTP) is far superior to that of GGEUR (FedVTP), leading by a margin of **2.53%** on CIFAR-100 ($\beta = 0.1$). This provides powerful empirical evidence that modeling the nonlinear manifold geometry is fundamentally more effective than relying on a flawed global linear approximation.

### 5.2.2 PERFORMANCE ON DOMAIN SKEW SCENARIOS

The superiority of FedMC is even more pronounced in the more challenging domain skew scenarios, achieving a remarkable **2.05%** lead on the difficult DomainNet dataset ($\beta = 0.1$), as detailed in Table 2. This is because methods based on prototype averaging or linear assumptions fail when confronted with distinct client manifolds (*e.g.*, photos vs. sketches). In contrast, FedMC's "geometry dictionary" learns these domain-specific geometric signatures, enabling a far more effective calibration and leading to a more generalizable prompt.

### 5.3 PART 2: FEDMC AS A GENERAL FL FRAMEWORK

To validate its universality, we apply FedMC to a diverse suite of seven foundational FL methods. The results, summarized in Table 3, show that FedMC provides consistent and significant performance gains across all algorithms, confirming its value as a general-purpose enhancement module.

The most critical evidence comes from the comparison with **FedProto**. FedProto is a strong baseline that already mitigates heterogeneity by sharing first-order statistics (prototypes). The significant additional gain provided by FedMC powerfully demonstrates that sharing statistics representing data posi-

Table 3: Generality of FedMC. Performance (% accuracy) on Office-Home-LDS ($\beta = 0.1$) across a broad suite of FL algorithms.

| FL Algorithm | Base | +GGEUR | +FedMC |
|---|---|---|---|
| FedAvg | 70.14 | 83.99 | **85.11** |
| SCAFFOLD | 74.82 | 83.96 | **85.25** |
| MOON | 76.83 | 78.08 | **80.73** |
| FedDyn | 65.99 | 84.09 | **86.32** |
| FedOPT | 65.59 | 84.20 | **86.58** |
| FedNTD | 75.46 | 82.46 | **84.91** |
| FedProto | 69.40 | 83.35 | **85.92** |

tion is insufficient. By capturing and leveraging higher-order geometric information that defines the manifold's shape, FedMC provides a more fundamental calibration signal. Across all methods,

the performance uplift from FedMC is consistently greater than that from GGEUR, reaffirming the universal superiority of the nonlinear manifold hypothesis.

## 6 CONCLUSION

This paper identifies and rectifies a core flaw in existing geometry-aware FL methods: their reliance on a flawed global linearity assumption. We propose **FedMC**, a novel framework that instead embraces a more realistic nonlinear manifold hypothesis, learning local data geometry and performing privacy-preserving, on-manifold calibration via a global geometry dictionary. Extensive experiments demonstrate that FedMC provides significant and universal performance gains across both state-of-the-art FPL and a wide range of classic FL algorithms, establishing a more faithful geometric foundation for tackling data heterogeneity in federated learning.

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

## A    APPENDIX

Here we provide a summary of the key notations used throughout the paper for easy reference.

Table 4: Summary of key notations.

| Symbol | Description |
|---|---|
| $K$ | Total number of clients. |
| $\mathcal{D}_k$ | The set of local data embeddings for client $k$. |
| $x, y$ | A data embedding and its corresponding label. |
| $x'$ | A calibrated (augmented) data embedding. |
| $\mathcal{M}$ | The underlying low-dimensional data manifold. |
| $E_{vision}(\cdot)$ | The frozen vision encoder (e.g., from CLIP). |
| $P_r$ | The global prompt model at communication round $r$. |
| **FedMC Framework Components** | |
| $\mathcal{B}_g$ | The Global Anonymous Basis (GAB), a set of shared reference points $\{b_s\}$. |
| $N_{\text{base}}$ | The number of basis vectors in the GAB. |
| $k(x, y)$ | The Gaussian (RBF) kernel function. |
| $\Phi(x)$ | The mapping to the high-dimensional feature space $\mathcal{H}$. |
| $\boldsymbol{\alpha}_{j,i}$ | Coefficient vector for the $i$-th principal component of local cluster $j$. |
| $\boldsymbol{\beta}_{j,i}$ | The secure representation of a principal component, projected onto the GAB. |
| LGD | Local Geometry Descriptor, uploaded by the client. |
| $\mathcal{D}_r$ | The global Geometry Dictionary at round $r$. |

## B    EXPERIMENTAL DETAILS

This appendix provides a detailed description of the datasets and implementation settings used in our experiments.

### B.1    DATASET DETAILS

Our dataset selection is consistent with recent state-of-the-art benchmarks Qiu et al. (2024) and covers three primary types of data heterogeneity.

**Label Skew.**    We evaluate performance on two single-domain image classification datasets where clients have imbalanced class distributions:

- **CIFAR-100** Krizhevsky et al. (2009): Contains 100 classes with 50,000 training and 10,000 validation images.
- **Tiny-ImageNet** Deng et al. (2009): A subset of ImageNet with 200 classes and 100,000 training images.

*Heterogeneity Setting:* Following Qiu et al. (2024), we partition the data among 10 clients using a Dirichlet distribution, $Dir(\beta)$. We experiment with $\beta \in \{0.5, 0.3, 0.1\}$, where a smaller $\beta$ value indicates a higher degree of label skew.

**Domain Skew.**    We use three standard multi-domain datasets where each client's data comes from a different feature distribution (domain):

- **Office-Home** Venkateswara et al. (2017): Comprises 65 classes across 4 domains: Art (A), Clipart (C), Product (P), and Real World (R).
- **Office-31** Gong et al. (2012): Consists of 31 classes from 3 domains: Amazon (A), DSLR (D), and Webcam (W).
- **DomainNet** Peng et al. (2019): A large-scale dataset with 345 classes across 6 domains: Sketch, Real, Painting, Clipart, Infograph, and Quickdraw.

*Heterogeneity Setting:* Consistent with Qiu et al. (2024), each domain is assigned to a distinct set of clients. The total number of clients is set to be twice the number of domains (e.g., $K = 8$ for Office-Home).

**Label and Domain Skew.** To evaluate performance in a more realistic and challenging setting where both label and domain shifts coexist, we employ the **Office-Home-LDS** Ma et al. (2025a). This dataset is constructed from Office-Home. While ensuring that each client is assigned data from only one domain, the number of samples per class on each client is determined by a coefficient matrix generated via a Dirichlet distribution with parameter $\beta$. This setup creates a complex scenario with simultaneous domain and label heterogeneity.

## B.2 IMPLEMENTATION DETAILS

**Network Architecture.** For all FPL-related experiments (Part 1), we use the pre-trained **CLIP ViT-B/16** model as the backbone, consistent with Qiu et al. (2024). For the experiments in Part 2, we use standard architectures appropriate for the datasets (e.g., ResNet-18 for CIFAR-100).

**Federated Settings.** To ensure a fair comparison, all experiments are conducted with the following unified federated hyperparameters: communication rounds $T = 50$, local epochs $E = 1$, SGD optimizer with momentum 0.9, batch size 32, and an initial learning rate of 0.001.

**FedMC Parameters.** For our FedMC framework, the size of the Global Anonymous Basis (GAB) is set to $N_{\text{base}} = 512$. The number of local clusters per client is $m = 3$. For Kernel PCA (KPCA), we employ the Gaussian (RBF) kernel. Its bandwidth parameter was set to a standard and effective default, $\gamma = 1/d$, where $d$ is the dimensionality of the feature embeddings (e.g., $d = 512$ for ViT-B/16).

**Evaluation Metric.** The performance of all methods is measured by the **average classification accuracy** on a global, unified test set, evaluated after the final communication round.

## B.3 SCALABILITY AND COMPUTATIONAL OVERHEAD ANALYSIS

Kernel-based manifold learning methods are often criticized for poor scalability with large datasets. However, in the context of Federated Learning (FL), **FedMC** is carefully designed to mitigate this issue through localized computation and structural constraints, ensuring its practicality even in large-scale settings.

**Local Computation is Bounded and Amortized.** All kernel-dependent operations (K-Means clustering, Kernel PCA, and pre-image reconstruction) are performed *locally* on each client and only *once per communication round*, not per local training step. Crucially, instead of applying Kernel PCA to the entire local dataset $\mathcal{D}_k$, we first partition it into $m$ small clusters (e.g., $m = 3$ in our experiments). Since Kernel PCA has $\mathcal{O}(n^3)$ complexity in the number of samples $n$, this clustering step effectively caps the input size for each KPCA instance, rendering the per-client cost manageable even as $|\mathcal{D}_k|$ grows.

**Empirical Time Overhead is Modest.** To quantify the practical overhead, we measure the average training time per communication round for FedVTP with and without FedMC on three large domain-skew datasets. As shown in Table 5, FedMC introduces only a small time increase—ranging from 2.8s to 4.1s per round—while delivering substantial accuracy gains (cf. Table 2).

Table 5: Average training time (seconds) per communication round.

| Method | Office-31 | Office-Home | DomainNet |
|---|---|---|---|
| FedVTP | 38.4 | 40.8 | 34.5 |
| FedMC (FedVTP) | 41.2 | 44.5 | 38.6 |

**Scalability with Respect to Number of Clients.** The server-side operations are highly scalable. The Global Anonymous Basis (GAB) is constructed once offline. In each round, geometry fusion

involves only weighted averaging of vectors (Eq. 7), which scales linearly with the number of participating clients. To validate robustness under massive client settings, we evaluate FedMC on CIFAR-100 ($\beta = 0.1$) with $K = 100$, 300, and 500 clients. As shown in Table 6, FedMC consistently outperforms the base FedVTP across all scales, confirming its resilience to client growth.

Table 6: Accuracy (%) on CIFAR-100 ($\beta = 0.1$) with varying numbers of clients $K$.

| Method | $K = 100$ | $K = 300$ | $K = 500$ |
|---|---|---|---|
| FedVTP | 75.73 | 73.51 | 69.85 |
| FedMC (FedVTP) | **77.95** | **76.44** | **73.26** |

**Computation vs. Communication Trade-off.** In FL, communication is typically the dominant bottleneck. By investing modest additional local computation to generate high-fidelity, on-manifold samples, FedMC enables faster convergence and higher final accuracy—often reducing the total number of rounds needed to reach a target performance. Thus, the slight increase in local compute is not only acceptable but strategically advantageous in bandwidth-constrained federated environments.

## C    DETAILED ALGORITHM AND PSEUDOCODE WALKTHROUGH

In this section, we provide the main algorithm for FedMC and detailed, commented pseudocode for its key components. This is intended to clarify the implementation steps and rationale behind our framework. First, we present the overall workflow of FedMC as applied to Federated Prompt Learning. Algorithm 1 shows the complete process, including the preparatory GAB construction and the iterative training loop.

### C.1    WALKTHROUGH OF KEY CLIENT-SIDE FUNCTIONS

To further demystify the core mechanisms of FedMC, we now provide a step-by-step walkthrough of the two most critical client-side functions, accompanied by Python-style pseudocode.

### C.1.1    GENERATING A SECURE LOCAL GEOMETRY DESCRIPTOR (LGD)

The primary challenge in sharing geometric information is privacy. Our solution, detailed in Section 4.2.1, is to transform locally computed geometry into a secure, standardized format using the shared GAB. This process involves three main steps:

**1. Local Geometry Extraction:**    The client first partitions its local data embeddings into small clusters, each assumed to be a relatively flat "patch" on the manifold. On each patch, it performs Kernel PCA to find the principal directions of variance in a high-dimensional feature space. This captures the local, non-linear structure.

**2. Secure Projection:**    This is the crucial privacy-preserving step. Instead of sharing the principal components (which depend on private data), the client projects them onto the public GAB. As shown in Eq. 5 and the code below, this results in a coefficient vector 'betas' which describes the local geometry purely in terms of the anonymous basis vectors. This representation is now safe to share.

**3. Assembling the LGD:**    Finally, the client packages this secure geometric information (the 'betas'), along with variance information ('lambdas') and other metadata, into a Local Geometry Descriptor (LGD) for upload.

The entire process is illustrated in the pseudocode below.

### C.1.2    PERFORMING MANIFOLD-AWARE CALIBRATION

Once a client receives the global Geometry Dictionary, it can perform on-manifold calibration for its local embeddings, as described in Section 4.3. This process effectively generates high-quality, synthetic data points that lie on the learned global manifold.

---

**Algorithm 1** FedMC for Federated Prompt Learning

---

1: **Server Initializes:** Pre-trained frozen encoders $E_{vision}, E_{text}$.
2: **// Preparatory Phase: GAB Construction**
3: Server requests initial prototypes from all $K$ clients.
4: **for** each client $k \in \{1, \ldots, K\}$ **in parallel do**
5:     Extract embeddings $\mathcal{D}_k = \{E_{vision}(\text{img}) | \text{img} \in \text{LocalImages}_k\}$.
6:     Compute local prototypes $\{c_{k,j}\}$ from $\mathcal{D}_k$.
7:     Anonymize and send prototypes $\{\tilde{c}_{k,j}\}$ to server.
8: **end for**
9: Server aggregates prototypes and computes GAB, $\mathcal{B}_g$.
10: Server initializes global prompt $P_0$ and distributes it with $\mathcal{B}_g$ to all clients.
11: Initialize Geometry Dictionary $\mathcal{D}_0 \leftarrow \emptyset$.

12: **// Federated Training Loop**
13: **for** round $r = 0, 1, \ldots, R - 1$ **do**
14:     Server selects a subset of clients $S_r$.
15:     **Server broadcasts** current prompt $P_r$ and dictionary $\mathcal{D}_r$ to clients in $S_r$.
16:     Let $\text{LGD}_{\text{all}} \leftarrow \emptyset$.
17:     **for** each client $k \in S_r$ **in parallel do**
18:         $P_{r+1}^k, \text{LGD}_k \leftarrow \text{ClientPromptUpdate}(k, P_r, \mathcal{D}_r, \mathcal{B}_g)$.
19:         Client $k$ sends updated prompt $P_{r+1}^k$ and $\text{LGD}_k$ to the server.
20:         $\text{LGD}_{\text{all}} \leftarrow \text{LGD}_{\text{all}} \cup \text{LGD}_k$.
21:     **end for**
22:     **// Server-Side Aggregation**
23:     Update global prompt: $P_{r+1} \leftarrow \text{Aggregate}(\{P_{r+1}^k\}_{k \in S_r})$.          $\triangleright$ e.g., FedAvg
24:     Fuse geometries: $\mathcal{D}_{r+1} \leftarrow \text{FuseGeometries}(\text{LGD}_{\text{all}})$. $\triangleright$ Meta-cluster weighted avg (Eq. 7)
25: **end for**

26: **function** CLIENTPROMPTUPDATE$(k, P_r, \mathcal{D}_r, \mathcal{B}_g)$
27:     Let local prompt $P_k \leftarrow P_r$.
28:     Extract embeddings from local images: $\mathcal{D}_k = \{E_{vision}(\text{img}) | \text{img} \in \text{LocalImages}_k\}$.
29:     **// Phase II: Manifold-Aware Calibration**
30:     $\hat{\mathcal{D}}_k \leftarrow \emptyset$.
31:     **for** each embedding-label pair $(x, y) \in \mathcal{D}_k$ **do**
32:         Generate calibrated embedding $x'$ from $x$ using $\mathcal{D}_r$ and $\mathcal{B}_g$.      $\triangleright$ Eqs. 9-13
33:         $\hat{\mathcal{D}}_k \leftarrow \hat{\mathcal{D}}_k \cup \{(x', y)\}$.
34:     **end for**
35:     **// Local Prompt Training**
36:     Train local prompt $P_k$ on calibrated dataset $\hat{\mathcal{D}}_k$ for $E$ epochs to get $P_{r+1}^k$.
37:     **// Phase I: Local Geometry Extraction**
38:     Extract and securely represent local geometry from $\mathcal{D}_k$ to get $\text{LGD}_k$.     $\triangleright$ Eq. 5
39:     **return** $P_{r+1}^k, \text{LGD}_k$.
40: **end function**

---

**1. Dynamic Geometry Query:** For any given embedding 'x', the client first "looks up" the most relevant geometric map in the dictionary by finding the closest prototype. This ensures the calibration is context-aware and uses the appropriate local geometry.

**2. On-Manifold Perturbation:** The client projects 'x' onto the principal components of the retrieved geometric template. It then adds a scaled random noise to these projection coordinates. This is analogous to taking a small, random step, but crucially, this step is constrained to the directions that are "valid" on the manifold (i.e., the tangent space).

**3. Pre-image Reconstruction:** The perturbed coordinates now exist in the high-dimensional feature space. The final, challenging step is to find the corresponding point 'x'' back in the original embedding space. This is a classic inverse problem which we solve via optimization, seeking an 'x'' whose feature-space representation is closest to our perturbed target.

```python
1  # Python-style Pseudocode for Secure Geometry Representation
2
3  def extract_secure_local_geometry(local_embeddings, GAB, n_clusters,
       n_components):
4      """
5      Client-side function to extract local geometry and represent it
       securely.
6      Corresponds to Section 4.2.1.
7      """
8      # local_embeddings: Embeddings from the client's private data
9      # GAB: Global Anonymous Basis received from the server
10
11     # Partition data into local patches
12     clusters = KMeans(n_clusters).fit_predict(local_embeddings)
13
14     client_LGDs = []
15     for cluster_id in range(n_clusters):
16         cluster_points = local_embeddings[clusters == cluster_id]
17
18         # 1. Local Geometry Extraction via Kernel PCA
19         kpca = KernelPCA(n_components=n_components, kernel="rbf")
20         kpca.fit(cluster_points)
21
22         # lambdas are the eigenvalues (variance)
23         # alphas are the coefficients in the dual space
24         lambdas = kpca.eigenvalues_
25         alphas = kpca.dual_coef_
26
27         # 2. Secure Projection onto GAB (The key privacy step - Eq. 5)
28         # This converts private geometry (defined by alphas and
       cluster_points)
29         # into a public representation (betas) using the GAB.
30         kernel_matrix = rbf_kernel(cluster_points, GAB) # k(x_a, b_s)
31         betas = alphas.T @ kernel_matrix # Shape: (n_components, N_base)
32
33         # 3. Assemble the Local Geometry Descriptor (LGD)
34         descriptor = {
35             "center_prototype": np.mean(cluster_points, axis=0),
36             "num_samples": len(cluster_points),
37             "geometry": [(lam, beta) for lam, beta in zip(lambdas, betas)
       ]
38         }
39         client_LGDs.append(descriptor)
40
41     return client_LGDs
```

Listing 1: Python-style pseudocode for generating Local Geometry Descriptors (LGDs) on the client side. This illustrates the core process of local KPCA followed by a secure projection onto the GAB.

This calibration pipeline is shown in the pseudocode below.

## C.2 Justification for Server-Side Geometric Fusion

In Section 4.2.2 (Eq. 7) of the main paper, we proposed a weighted averaging scheme for the server to fuse local geometric information from multiple clients into a single, representative template. Here, we provide a theoretical justification for this scheme by demonstrating that it is the optimal solution to a well-posed optimization problem.

**Problem Formulation.** Consider a meta-cluster $l$ on the server, which groups together a set of Local Geometry Descriptors (LGDs) describing the same region of the manifold. For each principal component index $i$, the server receives a set of secure representations $\{\boldsymbol{\beta}_{j,i}\}_{j \in l}$ from the clients in

```python
1  # Python-style Pseudocode for Manifold-Aware Calibration
2
3  def calibrate_embedding(x, geometry_dictionary, GAB):
4      """
5      Client-side function to perform on-manifold calibration for an
       embedding x.
6      Corresponds to Section 4.3.
7      """
8      # x: A single embedding vector to be calibrated
9      # geometry_dictionary: The global atlas received from the server
10
11     # 1. Dynamic Geometry Query (Eq. 8)
12     # Find the most relevant geometric template from the dictionary
13     prototypes = [entry["prototype"] for entry in geometry_dictionary]
14     best_entry_idx = find_nearest(x, prototypes)
15     template = geometry_dictionary[best_entry_idx]
16
17     fused_lambdas = template["lambdas"]
18     fused_betas = template["betas"] # Shape: (n_components, N_base)
19
20     # 2. Projection onto Principal Components (Eq. 9)
21     # Project x onto the local tangent space defined by the template
22     kernel_vec = rbf_kernel(x.reshape(1, -1), GAB) # k(x, b_s)
23     projections = fused_betas @ kernel_vec.T # p_i = sum_s (beta_is * k(x
       , b_s))
24
25     # 3. Perturbation in the Principal Component Space (Eq. 10)
26     noise = np.random.randn(len(fused_lambdas))
27     perturbed_projections = projections + noise * np.sqrt(fused_lambdas)
28
29     # 4. Reconstruction of the Perturbed Feature Vector (Pre-image
       problem)
30     # We need to find x' such that its projection matches the perturbed
       one.
31     # This step is an optimization problem (Eq. 11-13).
32     x_calibrated = solve_pre_image(perturbed_projections, fused_betas,
       GAB, init=x)
33
34     return x_calibrated
```

Listing 2: Python-style pseudocode for the client-side manifold-aware calibration process. It details how a client uses the global geometry dictionary to calibrate a single embedding.

this group. The goal is to find a single consensus vector $\boldsymbol{\beta}_{l,i}^*$ that best represents this collection of vectors.

We can formalize this as finding a vector $\boldsymbol{\beta}$ that minimizes the weighted sum of squared Euclidean distances to all client-provided vectors. This can be expressed as a weighted least squares problem:

$$\boldsymbol{\beta}_{l,i}^* = \arg \min_{\boldsymbol{\beta} \in \mathbb{R}^{N_{\text{base}}}} \sum_{j \in l} w_j \|\boldsymbol{\beta} - \boldsymbol{\beta}_{j,i}\|_2^2 \tag{15}$$

**Justification of the Weights.** The choice of weights $w_j$ is critical and should reflect the "confidence" or "importance" of the geometric information provided by client $j$. We argue that a suitable weight should be proportional to two factors:

- **Sample Size** ($n_j$): LGDs derived from a larger number of local data points ($n_j$) are statistically more stable and less prone to sampling noise.
- **Variance** ($\lambda_{j,i}$): The eigenvalue $\lambda_{j,i}$ represents the variance captured by the $i$-th principal component. A larger variance indicates that this direction is more significant for describing the local data structure.

Therefore, a natural and effective choice for the weight is $w_j = n_j \lambda_{j,i}$.

**Derivation of the Solution.** The objective function in Eq. 15, let's call it $L(\boldsymbol{\beta})$, is convex and differentiable with respect to $\boldsymbol{\beta}$. To find the minimum, we can set its gradient to zero:

$$\nabla_{\boldsymbol{\beta}} L(\boldsymbol{\beta}) = \nabla_{\boldsymbol{\beta}} \sum_{j \in l} w_j (\boldsymbol{\beta} - \boldsymbol{\beta}_{j,i})^T (\boldsymbol{\beta} - \boldsymbol{\beta}_{j,i}) \tag{16}$$

$$= \sum_{j \in l} 2 w_j (\boldsymbol{\beta} - \boldsymbol{\beta}_{j,i}) \tag{17}$$

Setting the gradient to zero to find the optimal $\boldsymbol{\beta}_{l,i}^*$:

$$\sum_{j \in l} 2 w_j (\boldsymbol{\beta}_{l,i}^* - \boldsymbol{\beta}_{j,i}) = 0 \tag{18}$$

$$\left( \sum_{j \in l} w_j \right) \boldsymbol{\beta}_{l,i}^* = \sum_{j \in l} w_j \boldsymbol{\beta}_{j,i} \tag{19}$$

$$\boldsymbol{\beta}_{l,i}^* = \frac{\sum_{j \in l} w_j \boldsymbol{\beta}_{j,i}}{\sum_{j \in l} w_j} \tag{20}$$

By substituting our chosen weight $w_j = n_j \lambda_{j,i}$, we arrive at the exact fusion formula for the coefficient vectors as presented in Eq. 7:

$$\boldsymbol{\beta}_{l,i}^* = \frac{\sum_{j \in l} n_j \lambda_{j,i} \boldsymbol{\beta}_{j,i}}{\sum_{j \in l} n_j \lambda_{j,i}} \tag{21}$$

This derivation shows that our intuitive weighted averaging scheme is not ad-hoc, but is in fact the optimal solution for finding a geometric consensus under a weighted least squares criterion. The fusion of eigenvalues $\lambda_{l,i}^*$ follows a similar logic, averaging them based on the number of samples $n_j$ that contributed to each estimate.

### C.3 GRADIENT DERIVATION FOR PRE-IMAGE RECONSTRUCTION

In Section 4.3 and Appendix B.2.2, we described the process of finding a pre-image $x'$ by minimizing a loss function (Eq. 13). To solve this optimization problem using gradient-based methods (e.g., Gradient Descent or Adam), we must compute the gradient of the loss function $\mathcal{L}(x')$ with respect to the variable $x'$. We provide the detailed derivation below.

**The Loss Function.** The objective is to find an $x'$ whose kernel evaluations with the GAB vectors $\{b_s\}$ match the target values $\{T_s\}$ as closely as possible. The loss function is:

$$\mathcal{L}(x') = \sum_{s=1}^{N_{\text{base}}} \left( k(x', b_s) - T_s \right)^2 \tag{22}$$

where $k(\cdot, \cdot)$ is the Gaussian (RBF) kernel.

**Step 1: Applying the Chain Rule.** We first compute the gradient of $\mathcal{L}(x')$ with respect to $x'$ using the chain rule:

$$\nabla_{x'} \mathcal{L}(x') = \sum_{s=1}^{N_{\text{base}}} 2 \left( k(x', b_s) - T_s \right) \cdot \nabla_{x'} k(x', b_s) \tag{23}$$

**Step 2: Gradient of the RBF Kernel.** Next, we need the gradient of the RBF kernel function $k(z, c) = \exp(-\gamma \|z - c\|_2^2)$ with respect to its first argument $z$.

$$\nabla_z k(z, c) = \nabla_z \exp(-\gamma \|z - c\|_2^2) \tag{24}$$

$$= \exp(-\gamma \|z - c\|_2^2) \cdot \nabla_z (-\gamma \|z - c\|_2^2) \tag{25}$$

$$= k(z, c) \cdot (-\gamma) \cdot \nabla_z \left( \sum_d (z_d - c_d)^2 \right) \tag{26}$$

$$= k(z, c) \cdot (-\gamma) \cdot (2(z - c)) \tag{27}$$

$$= -2\gamma(z - c)k(z, c) \tag{28}$$

Applying this to our case, we have $\nabla_{x'} k(x', b_s) = -2\gamma(x' - b_s)k(x', b_s)$.

**Step 3: Final Gradient Expression.** Finally, by substituting the kernel gradient (Eq. 28) back into the chain rule expression (Eq. 23), we obtain the complete gradient for the loss function:

$$\nabla_{x'} \mathcal{L}(x') = \sum_{s=1}^{N_{\text{base}}} 2 \left( k(x', b_s) - T_s \right) \cdot \left( -2\gamma(x' - b_s)k(x', b_s) \right) \tag{29}$$

$$= -4\gamma \sum_{s=1}^{N_{\text{base}}} \left( k(x', b_s) - T_s \right) k(x', b_s)(x' - b_s) \tag{30}$$

This final gradient expression can be directly implemented and used in any standard iterative optimization algorithm to solve for the pre-image $x'$. For instance, the update rule for gradient descent would be $x'_{t+1} = x'_t - \eta \nabla_{x'} \mathcal{L}(x'_t)$, where $\eta$ is the learning rate.

