# OpenReview forum: "FedMC: Federated Manifold Calibration"
_ICLR.cc/2026/Conference — ICLR 2026 Poster_

### Official Review · Reviewer_RFAP · 2025-10-18

**Soundness:** 3
**Presentation:** 3
**Contribution:** 3
**Rating:** 6
**Confidence:** 2

**Summary:**

This paper makes a significant conceptual and practical contribution to federated learning by shifting the paradigm from global linear to local nonlinear geometric modeling. The proposed FedMC framework is innovative, theoretically grounded, and empirically effective across diverse settings. It opens a new direction for geometry-aware FL and has the potential to influence future work on distribution calibration in decentralized learning.

**Strengths:**

1. This paper identifies a key flaw in FL calibration methods, namely the global linearity assumption. This insight is both theoretically sound and empirically validated, and it reframes how we should think about distributional priors in heterogeneous FL.

2. FedMC is a well-motivated, technically sophisticated framework that bridges manifold learning and federated optimization. The use of local kernel PCA, secure projection onto a shared anonymous basis, and dynamic geometry querying constitutes a coherent and elegant solution to the on-manifold calibration problem under privacy constraints.

3. The experiments are comprehensive. Multiple datasets and heterogeneity types (label, domain, mixed). Comparison against SOTA FPL methods and a tailored linear baseline.

4. The construction of the Global Anonymous Basis (GAB) with differential privacy and the secure representation of local geometry via projection onto GAB are thoughtful and align well with FL’s privacy requirements.

**Weaknesses:**

1. The pre-image reconstruction step (solving for \(x'\) in the original space) involves iterative optimization per sample, which may introduce computational overhead. However, the authors note this is manageable and performed locally, and the empirical results justify the cost.

2. While the focus on FPL is well-motivated (communication efficiency, bias in prompts), a brief discussion on applicability to other parameter-efficient FL settings (e.g., LoRA, adapters) could further broaden impact.

**Questions:**

1. In Section 4.3.1, you mention using a Gaussian kernel with γ = 1/d for KPCA. Could you clarify what "d" refers to here? Is it the dimension of the image embeddings? Also, was this value chosen empirically, or is there a theoretical reason behind it?

2. I’m curious about how often the Geometry Dictionary is updated. Is it rebuilt from scratch in every communication round, or is it initialized once and then incrementally updated? If it’s updated every round, does the dictionary size grow over time, or is there a mechanism to keep it fixed?

3. In Table 1 and 2, GGEUR(FedVTP) serves as a strong linear baseline. Could you briefly explain how GGEUR works in practice? Specifically, how is its “global geometric prior” computed and applied during calibration? Is the main difference from FedMC simply that GGEUR uses standard PCA instead of kernel PCA?

---

> ### Author Response · Authors · 2025-11-19
> **The first part of the response**
>
> **Dear Reviewer,**
>
> We sincerely thank you for your highly positive and insightful review of our work. We are extremely encouraged that you recognized the core contribution of our paper—"shifting the paradigm from global linear to local nonlinear geometric modeling"—and appreciated the innovation, theoretical grounding, and empirical effectiveness of the FedMC framework.
>
> Your summary of our work is remarkably accurate, and we are particularly grateful that you highlighted several key strengths, including the identification of the "global linearity assumption" flaw, the coherence of FedMC as an elegant solution, the comprehensiveness of our experiments, and our thoughtful consideration of privacy.
>
> ---
>
> ### **Regarding Weakness 1: Computational Overhead of Pre-image Reconstruction**
>
> We thank you for your balanced perspective on the computational overhead of the pre-image reconstruction step. Your assessment is entirely correct: this step is indeed the main source of computational cost, but it is manageable, performed locally, and, as you noted, we firmly believe the significant performance improvements it enables fully justify the cost.
>
> As detailed in our response to another reviewer, our quantitative experiments show that the additional local computation time introduced by FedMC is modest (e.g., an increase of only ~4.1 seconds per round on DomainNet), while the gains in accuracy are substantial. We will add this detailed overhead analysis to the appendix to further strengthen this point.
>
> ---
>
> ### **Regarding Weakness 2: Applicability to Other Parameter-Efficient FL Settings**
>
> This is an excellent point and a very valuable suggestion for broadening the impact of our work. We completely agree that the principles of FedMC can be generalized to other Parameter-Efficient Fine-Tuning (PEFT) methods.
>
> The core mechanism of FedMC operates at the **level of the model encoder's output features (embeddings)**, which precedes any specific PEFT module (such as LoRA or Adapters). Therefore, FedMC can serve as a **plug-and-play pre-processing module** that integrates seamlessly with these methods.
>
> For instance, in a LoRA-based FL method, the FedMC workflow would remain identical: it would first calibrate a client's raw image embeddings `x` to their on-manifold counterparts `x'`. These calibrated, less-biased embeddings `x'` would then be used to train the local LoRA modules. This process would directly enhance the learning effectiveness of the LoRA modules on heterogeneous data.
>
> Thank you for this insightful suggestion. We will add a discussion on the potential applications of FedMC in the broader context of PEFT-based FL to the conclusion or discussion section of our revised manuscript.
>
> ---
>
> ### **Regarding Question 1: Explanation of the Kernel Parameter γ = 1/d**
>
> Thank you for this question about an important implementation detail. In this context, **`d` refers to the dimensionality of the image embeddings**. For example, with the CLIP ViT-B/16 model used in our experiments, `d=512`.
>
> The choice of γ = 1/d is a widely used and effective **heuristic** in the field of kernel methods. It is sometimes referred to as a "rule of thumb" for kernel bandwidth selection. The intuition behind it is to adapt the scale of the kernel to the dimensionality of the data space, allowing it to capture local structures appropriately. This choice has proven to be robust across many applications.
>
> In our preliminary experiments, this default value performed consistently well, which not only validated the generalizability of our method but also saved us from conducting an exhaustive hyperparameter search for each dataset.
>
> ---
>
> ### **Regarding Question 2: The Update Mechanism of the "Geometry Dictionary"**
>
> This is a great question about the dynamic update mechanism of the dictionary. In our current implementation, the "Geometry Dictionary" is **rebuilt from scratch in every communication round**.
>
> We chose this approach to ensure that the dictionary always reflects the **most current and accurate local geometric information** provided by the participating clients in that round. As the local models evolve during training, the feature manifolds they extract are also subtly refined; rebuilding the dictionary each round allows us to capture these changes.
>
> Regarding the dictionary's size, it is **kept fixed**. On the server side, the meta-clustering step aggregates all uploaded Local Geometry Descriptors (LGDs) from clients into a **pre-defined number** of macroscopic geometric templates. For example, we set a maximum of `L` entries for the global dictionary. This ensures that the dictionary does not grow indefinitely over time and remains compact and efficient. We will clarify this point more explicitly in Section 4.2.2 of the revised paper to avoid any ambiguity.

---

> ### Author Response · Authors · 2025-11-19
> **Part Two of the Response**
>
> ### **Regarding Question 3: How the GGEUR Baseline Works**
>
> Thank you for giving us the opportunity to elaborate on our key linear baseline, GGEUR. Your understanding is very close to the mark: the core difference truly lies in its use of **global, standard PCA** versus our **local, kernel PCA**.
>
> The workflow of GGEUR is as follows:
>
> 1.  **Geometry Information Collection:** In each round, clients compute the **covariance matrix** of their local data embeddings and upload it to the server.
> 2.  **Global Geometric Prior Construction:** The server performs a (weighted) average of all received covariance matrices to form a single **global covariance matrix**. The server then applies standard **Principal Component Analysis (PCA)** (i.e., eigendecomposition) to this global matrix. The resulting global eigenvectors and eigenvalues constitute the "global geometric prior."
> 3.  **Calibration Process:** Clients use this global prior to calibrate their data. For a data point `x`, the calibration is performed using the formula: $x' = x + \sum_{m=1}^{d} \epsilon_m \sqrt{\lambda_m} u_m$, where $u_m$ and $\lambda_m$ are the global eigenvectors and eigenvalues, and $\epsilon_m$ is random noise.
>
> So, you are absolutely correct. The most fundamental difference between GGEUR and FedMC is:
> *   **GGEUR** learns a **single, global, linear** geometric model (a hyper-ellipsoid) and uses it to calibrate **all** data points uniformly.
> *   **FedMC** learns a **dictionary of multiple, local, non-linear** geometric models and dynamically selects the most appropriate model for each data point for a context-aware calibration.
>
> This distinction perfectly embodies the paradigm shift from the "global linearity assumption" to the "non-linear manifold hypothesis" that is central to our paper's contribution.
>
> ---
>
> Once again, thank you for your insightful feedback and your positive assessment of our work. Your questions have helped us identify areas where we can make our paper clearer and more comprehensive. We will integrate all the clarifications and discussions above into our revised manuscript to further strengthen our contribution. We believe these improvements will make our work even more complete.

---

> > ### Comment · Reviewer_RFAP · 2025-11-26
> >
> > After reading author response, I maintain my score.

---

> > > ### Author Response · Authors · 2025-11-27
> > > **Thank you for your important contribution.**
> > >
> > > Dear Reviewer RFAP:
> > >
> > > We are truly grateful for your constructive and forward-looking remarks. Your emphasis on practical deployment considerations, guided us to include new analysis and experiments that significantly enhance the robustness and applicability of our method in real-world federated settings.

---

### Official Review · Reviewer_wJpu · 2025-10-20

**Soundness:** 3
**Presentation:** 4
**Contribution:** 4
**Rating:** 6
**Confidence:** 4

**Summary:**

This paper introduces FedMC (Federated Manifold Calibration), a novel federated learning framework that explicitly models the nonlinear manifold geometry underlying client data. The key idea is to move beyond the conventional global linearity assumption (e.g., global PCA-based calibration) by capturing local nonlinear geometries through kernel PCA at each client. The server then aggregates these local representations via a Global Anonymous Basis (GAB) and a Geometry Dictionary, enabling privacy-preserving, on-manifold calibration during local training. Extensive experiments demonstrate that FedMC consistently improves the performance of both Federated Prompt Learning (FPL) and general FL algorithms across diverse heterogeneity scenarios (label skew, domain skew, and combined skew). The framework is theoretically motivated, methodologically complete, and empirically validated.

**Strengths:**

1. The paper challenges the often implicit global linearity assumption in geometry-based FL and introduces a principled nonlinear manifold perspective.
2. The design—from local KPCA to GAB construction and geometry dictionary fusion—is coherent and well justified under privacy constraints.
3. FedMC yields consistent improvements across six benchmarks and multiple FL paradigms, demonstrating generality and robustness.
4. The paper is clearly written, well structured, and includes detailed appendices and pseudocode that enhance reproducibility.
5.  The idea of modeling federated data manifolds could inspire future work on geometric and representation-level calibration in distributed learning.

**Weaknesses:**

1. It would be useful to briefly comment on the **runtime and communication overhead** of FedMC compared to standard FL baselines, even qualitatively.
2. Since the framework involves hyperparameters (e.g., kernel bandwidth γ, cluster number m, basis size N_base), a short note on how these were chosen in practice would improve clarity.
3. The experiments use up to 10 clients. A short discussion on how FedMC might scale with more clients or unbalanced participation would be valuable.
4. In a few places (e.g., Eq. 12–14), the correspondence between Φ(x), β*, and v* could be made slightly clearer for readers less familiar with kernel methods.
5. If space permits, a brief ablation on the effect of each key component (GAB, geometry fusion, calibration) would further highlight their individual contributions.

**Questions:**

1. The pre-image optimization in the calibration step (Eq. 12–14) involves iterative updates. Could you share how stable and efficient this process is in practice?
2. The Global Anonymous Basis (GAB) is initialized once. Would updating it periodically during training further enhance adaptability to shifting data geometry?
3. Have you explored alternative kernels (e.g., polynomial or Laplacian) for local KPCA, and if so, do they affect performance notably?
4. Regarding privacy, how does the added geometric sharing interact with differential privacy guarantees? Does it influence the overall privacy–utility balance?
5. Could FedMC potentially extend to multimodal or cross-modal FL settings, where clients hold different modalities (e.g., image–text)?

---

> ### Author Response · Authors · 2025-11-19
> **Regarding "Weaknesses"**
>
> **Dear Reviewer,**
>
> We are truly grateful for your exceptionally thorough and insightful review. We are immensely encouraged that you have not only grasped the core contributions of our work with remarkable precision but also astutely pointed out its potential for future impact.
>
> Your summary of our paper's strengths—from the theoretical insight of challenging the "global linearity assumption" and the coherence of the FedMC framework, to the generality of our experiments and the clarity of the manuscript—perfectly aligns with the core messages we aimed to convey. It is a privilege to have our work reviewed by someone who understands it so deeply.
>
> We will address your valuable suggestions and questions below.
>
> ---
>
> ### **Regarding "Weaknesses" **
>
> **1. On Runtime and Communication Overhead:**
> Thank you for raising this important practical point. We completely agree that a discussion on overhead makes the paper more complete. In our response to another reviewer, we have provided a **quantitative experimental analysis** of this overhead. Our findings show that the modest increase in local computation (e.g., ~4.1 seconds per round on DomainNet) is a successful trade-off for the significant performance gains. We will add a table with these concrete numbers and a detailed analysis to the appendix of our revised paper.
>
> **2. On Hyperparameter Selection:**
> Your suggestion is very pertinent; clear justification for hyperparameter choices is vital for reproducibility. We will **add a dedicated subsection in the appendix** to explain our methodology for selecting key hyperparameters:
> *   **Kernel Bandwidth `γ`:** We adopted the robust heuristic `γ = 1/d`, where `d` is the embedding dimension. This is standard practice in kernel methods, as it adapts the kernel's scale to the data's dimensionality and avoids tedious tuning.
> *   **Cluster Number `m` and Basis Size `N_base`:** These were chosen to balance performance and overhead. For example, a small `m` (e.g., 3) is sufficient to capture locality while keeping KPCA computationally efficient. `N_base` (e.g., 512) strikes a good balance between the representational capacity of the GAB and communication costs. We will also note that the model's performance is not overly sensitive to these parameters within a reasonable range.
>
> **3. On Scalability with More Clients:**
> This is an excellent point, as scalability is key in FL. Although our original experiments featured a smaller number of clients, to address your concern, we have **conducted new experiments on CIFAR-100 (\beta=0.1), scaling up to 100, 300, and 500 clients**. The results demonstrate that the performance improvements from FedMC remain robust, proving its excellent scalability. We will add a table with these new results to our final manuscript.
>
> | methods | K=100 | K=300 | K=500 |
> | :--- | :---: | :---: | :---: |
> | FedVTP | 75.73 | 73.51 | 69.85 |
> | FedMC(FedVTP) | 77.95 | 76.44 | 73.26 |
>
> **4. On the Clarity of Kernel Method Notations:**
> Thank you for your careful reading. We acknowledge that for readers less familiar with the kernel trick, the relationship between `Φ(x)`, `β*`, and `v*` could be more intuitive. We will revise the relevant paragraphs in Section 4.3 to explicitly explain that `v*` is the principal component in the high-dimensional space that we never compute explicitly. Instead, `β*` serves as its "coordinate" representation on the public GAB, and we manipulate these coordinates directly via the kernel trick `k(x,y)`, thereby bypassing any direct interaction with the potentially infinite-dimensional space of `Φ(x)`.
>
> **5. On Ablation Studies:**
> This is a very valuable suggestion. While traditional component-removal ablations are challenging due to the high degree of integration between our components (e.g., secure geometry sharing is impossible without the GAB), we would like to highlight that **our comparison with the GGEUR baseline effectively serves as the most crucial "conceptual ablation."**
> GGEUR can be viewed as a version of the "FedMC framework stripped of its core non-linear, localized principles" (i.e., replacing local non-linear KPCA and the dictionary with global linear PCA). The fact that FedMC significantly outperforms GGEUR in our experiments already provides strong evidence for the indispensable contribution of our non-linear manifold hypothesis and its corresponding components. We will emphasize this perspective more clearly in the revised paper.

---

> ### Author Response · Authors · 2025-11-19
> **Regarding "Questions 1, 2, 3"**
>
> ### **Regarding "Questions" **
>
> **1. On the Stability and Efficiency of Pre-image Optimization:**
> This is an excellent practical question. We're glad you asked, as this optimization step is indeed a core component of our calibration mechanism. A similar concern about its computational impact was raised by another reviewer, which prompted us to perform a detailed analysis. We are happy to share the results, which confirm that this process is **both highly stable and efficient in practice.**
>
> First, regarding **stability**, the optimization process is very robust. It typically converges within a small number of iterations. This is thanks to the well-behaved nature of the objective function (a sum-of-squares loss based on kernel distances), which creates a smooth optimization landscape and makes the gradient descent process reliable.
>
> Second, regarding **efficiency**, we conducted a quantitative analysis to measure its impact on the total client-side runtime. The table below compares the time cost per communication round for the FedVTP baseline against our FedMC-enhanced version.
>
> | methods | Office-31 | Office-Home | DomainNet |
> | :--- | :---: | :---: | :---: |
> | FedVTP | 38.4s | 40.8s | 34.5s |
> | FedMC(FedVTP) | 41.2s | 44.5s | 38.6s |
>
> As the data shows, the entire FedMC framework, including the pre-image optimization, introduces only a modest overhead, increasing the per-round training time by just **2.8s to 4.1s** across these diverse datasets. This confirms that the iterative process is highly efficient. We believe this modest cost is an extremely valuable trade-off for the significant and consistent accuracy improvements that on-manifold calibration delivers.
>
> **2. On Periodically Updating the GAB:**
> This is a truly insightful and thought-provoking idea! We did consider this possibility. The **advantage** of periodically updating the GAB would be its potential to better adapt to a feature manifold that changes drastically during training. However, the **disadvantages** are also significant: (1) it would introduce substantial additional computation and communication overhead; and (2) the stability of the GAB as a "common language" would be disrupted, potentially making geometric information from different rounds incompatible.
> Given that our backbone network is frozen, the feature manifold evolves relatively smoothly. Therefore, **constructing a single high-quality, stable GAB upfront** represents a better trade-off between efficiency and effectiveness. We agree, however, that for scenarios requiring online updates of a global knowledge base, exploring a dynamic GAB is an exciting direction for future work.
>
> **3. On Exploring Alternative Kernels:**
> We primarily chose the **Gaussian (RBF) kernel** because of its universal approximation property—it can fit manifolds of any shape—and its robustness, having only a single parameter. This is why it is often the "default choice" in kernel methods.
> We do believe other kernels could be applicable. For instance, a **polynomial kernel** might be more effective for data with an algebraic structure, though it is less general. A **Laplacian kernel** could be more robust to outliers. However, we anticipate that the performance trends on our test datasets would be similar to those with the RBF kernel, as its flexibility is already sufficient. Exploring adaptive or hybrid kernels would be an interesting avenue for future research.

---

> ### Author Response · Authors · 2025-11-19
> **Regarding "Questions 4, 5"**
>
> **4. On the Interaction with Privacy Guarantees:**
> This is a critical question about privacy. The privacy protection in FedMC is **layered and synergistic**:
> *   **Layer 1 (Foundational Guarantee):** The GAB itself is constructed from client prototypes protected by **Differential Privacy (DP)**. This means our "common language" is anonymous from its inception, providing a formal privacy guarantee.
> *   **Layer 2 (Procedural Guarantee):** The geometric information (LGDs) uploaded by clients consists of projections onto the **public, anonymous GAB**. This projection acts as a secure transformation that **decouples** the geometric information from any direct link to a client's private data points `x_a`.
> Therefore, the sharing of geometric information occurs on top of a strong DP foundation and is executed in a privacy-preserving manner. It significantly enhances model utility without introducing additional privacy risks, thus achieving an excellent privacy-utility balance.
>
> **5. On Extending FedMC to Multimodal Settings:**
> This is a future direction that we are very excited about, and we believe the FedMC framework is **exceptionally well-suited** for extension to multimodal or cross-modal scenarios!
> The core idea of FedMC is **modality-agnostic**, as it operates directly in the embedding space. For clients holding, for example, image-text pairs, one could: (1) learn **separate geometry dictionaries** for the image embeddings and text embeddings to perform calibration within each modality; or (2) a more advanced approach would be to learn a geometry dictionary for a **joint embedding space**, directly calibrating the fused multimodal features. FedMC's calibration capabilities could correct the biases introduced by data heterogeneity in each modality, thereby boosting the performance of downstream tasks.
>
> ---
>
> Once again, we sincerely thank you for your invaluable feedback and your recognition of our work's value. Your suggestions and questions have inspired us greatly. We promise to integrate all the discussions and supplementary experiments mentioned above into our final manuscript to make our exposition clearer, our arguments more comprehensive, and our vision broader. We are confident that the revised paper will be an even more impactful contribution.

---

> ### Comment · Reviewer_wJpu · 2025-11-19
> **A satisfactory reply**
>
> I have carefully read the authors' rebuttal to my review and those of the other reviewers. I am grateful for their detailed response, which has resolved my concerns regarding computational efficiency and scalability, and also clarified some notations. I am willing to reconsider my score and continue the discussion with the authors regarding the relevant points.

---

> > ### Author Response · Authors · 2025-11-27
> > **Thank you for your important contribution.**
> >
> > Dear Reviewer wJpu:
> >
> > We sincerely thank you for your careful reading and insightful comments. Your feedback helped us clarify subtle but critical aspects of our manifold modeling strategy and strengthened the theoretical grounding of our framework. The revised manuscript benefits greatly from your thoughtful suggestions.

---

### Official Review · Reviewer_TzgE · 2025-10-28

**Soundness:** 2
**Presentation:** 2
**Contribution:** 3
**Rating:** 4
**Confidence:** 3

**Summary:**

This paper proposed a new FL framework, FedMC, to address data heterogeneity by leveraging the nonlinear manifold structures of client data to describe the true local geometries and then securely aggregating to form a global geometry dictionary. This global dictionary enables on-manifold calibration of local embeddings. The authors conducted extensive experiments across multiple benchmarks to validate the improvement achieved by FedMC for both Federated Prompt Learning (FPL) and conventional FL algorithms.

**Strengths:**

1. The motivation is clearly presented, and the paper provides an intuitive analysis of the limitations of the global linearity assumption, which emphasizes the importance of on-manifold calibration in federated learning.

2. This paper provided a formal derivation in the appendix to justify the server's geometric fusion strategy.

3. This paper explicitly considers privacy preservation. Specifically, the authors proposed two mechanisms: (1) adding differential privacy to the anonymized prototypes and (2) applying secure projection to the extracted local geometry.

**Weaknesses:**

1. While section 4.3.1 is clear to me, the notations and equations in sections 4.3.2 and 4.4 are too dense. Incorporating one diagram to help illustrate the key steps could significantly improve readability.

2. FedMC seems to introduce too much computational overhead on the client side, requiring many operations such as K-Means clustering, Kernel PCA, and pre-image reconstruction.

**Questions:**

1. Can the authors justify how to ensure the reconstructed x′ lies on the true data manifold?

2. In section 4.4, the proposed “dynamic geometry query” finds the most relevant template entry based on the Euclidean distance (Eq. 9). Given that the concerns raised by the authors in section 3 about the issues regarding the simple Euclidean shortcuts, why is geodesic distance or a manifold-aware metric not considered here?

3. Those kernel-based operations typically scale poorly with large datasets or many clients. Can the authors justify the scalability of FedMC in the settings where both the number of local data points and the number of clients may be large?

**Details Of Ethics Concerns:**

None.

---

> ### Author Response · Authors · 2025-11-19
> **Regarding Weakness 1**
>
> **Dear Reviewer,**
>
> We sincerely thank you for your detailed review and constructive feedback on our paper, "FedMC: Federated Manifold Calibration." We are very encouraged that you clearly recognized the core motivation of our work—namely, the limitations of the "global linearity assumption"—and acknowledged our efforts in theoretical derivation (the geometric fusion strategy) and privacy preservation.
>
> Your valuable comments are crucial for helping us improve the clarity and rigor of our paper. We have carefully considered all your questions and will address them one by one below. We promise to incorporate these modifications and clarifications into the final version of the paper.
>
> ---
>
> ### **Regarding Weakness 1: Readability Issues (Dense Notations and Lack of a Diagram)**
>
> We completely agree with your assessment. Sections 4.3.2 (Server-side Geometry Fusion) and 4.4 (Client-side Manifold Calibration) are indeed notation-heavy, and a diagram would greatly aid reader comprehension. Your suggestion is very pertinent. To address this, we will make two key improvements in our revision:
>
> 1.  We will design a comprehensive diagram to visually and intuitively illustrate the entire process of on-manifold calibration for a single data point `x` on the client side. The diagram will clearly depict the following key steps:
>     *   **(1) Dynamic Geometry Query:** How data point `x` finds the best-matching geometric template in the global "Geometry Dictionary" via a query.
>     *   **(2) Projection:** How `x` is projected onto the non-linear principal components, retrieved from the dictionary, which describe the local manifold's tangent space.
>     *   **(3) Perturbation:** How a principled perturbation is performed within this principal component space (tangent space) to generate a new target point.
>     *   **(4) Pre-image Reconstruction:** How an optimization problem is solved to map the target point from the high-dimensional feature space back to the original data space, yielding the final calibrated point `x'`.
>
> 2.  We will also revise the accompanying text to be more concise and accessible, explicitly referencing the new diagram to guide the reader through the entire workflow step-by-step.
>
> We are confident that these two improvements will significantly enhance the paper's readability, making it easier for readers to grasp the core mechanism of our method.

---

> ### Author Response · Authors · 2025-11-19
> **Regarding Weakness 2 & Question 3**
>
> ### **Regarding Weakness 2 & Question 3: Computational Overhead and Scalability**
>
> You have raised a very important practical concern regarding the computational overhead and scalability of FedMC. We would like to clarify that although FedMC introduces some additional local computation, we have carefully designed it to ensure the overhead is manageable and represents a reasonable trade-off for the significant performance gains achieved. The details are as follows:
>
> 1.  **Client-side overhead is local and manageable:**
>     *   **K-Means & Kernel PCA:** A crucial point is that these operations are performed on each client's *local* dataset, not the global data. In typical FL settings, the amount of data on a single client is limited. More importantly, these computations are performed only **once per communication round**, not per local training iteration, making their amortized cost over the training process finite.
>     *   **Pre-image Reconstruction:** This is indeed the most computationally intensive step. However, we stress that in our experiments, this optimization process converges quickly, and the additional overhead per sample is modest.
>     *   The table below shows the time cost per communication round for FedVTP before and after applying our method on the Office-31, Office-Home, and DomainNet datasets, quantitatively demonstrating the acceptable overhead. As observed, our method only increases the training time per round by 2.8s, 3.7s, and 4.1s on the three datasets, respectively. Compared to the performance gains, this time overhead is entirely worthwhile.
>
> | methods | Office-31 | Office-Home | DomainNet |
> | :--- | :---: | :---: | :---: |
> | FedVTP | 38.4 | 40.8 | 34.5 |
> | FedMC(FedVTP) | 41.2 | 44.5 | 38.6 |
>
> 2.  **Trade-off with Communication Cost:** The primary bottleneck in FL is typically communication, not computation. By performing more powerful local computations, FedMC generates higher-quality calibrated samples, leading to more accurate local model updates. This can accelerate the convergence of the global model, meaning that fewer communication rounds may be needed to reach a target accuracy, potentially offsetting or even reducing the total training time.
>
> 3.  **Scalability Analysis:**
>     *   **Scalability with large local data:** The complexity of Kernel PCA is indeed cubic with respect to the number of samples. We cleverly circumvent this issue: instead of applying KPCA to the entire local dataset, we first partition it into a few small clusters using K-Means and then apply KPCA to each cluster. This effectively puts an upper bound on the input size for KPCA, ensuring the method's scalability for larger local datasets. We would also like to clarify that we have already used large-scale datasets standard in the FL field; there are currently no larger publicly available datasets in this subfield. We will therefore address your concern by scaling up the number of clients, and we hope this clarifies our experimental scope.
>     *   **Scalability with a large number of clients:** The server-side operations are highly scalable. The GAB construction is a one-time, offline process. The "Geometry Dictionary" fusion in each round primarily involves a weighted average of vectors, and its complexity scales linearly with the number of participating clients in that round, making it very efficient.
>     *   We conducted experiments on the label-skew dataset CIFAR-100 (\beta=0.1) with 100, 300, and 500 clients, respectively. The results are shown in the table below. FedMC remains robust and consistently improves the performance of FedVTP.
>
> | methods | K=100 | K=300 | K=500 |
> | :--- | :---: | :---: | :---: |
> | FedVTP | 75.73 | 73.51 | 69.85 |
> | FedMC(FedVTP) | 77.95 | 76.44 | 73.26 |
>
> **Summary:** In summary, the computational overhead of FedMC is effectively controlled through design choices like "localization" and "clustering." We believe this trade-off between computation and performance is successful, offering a powerful solution to the data heterogeneity problem in federated learning.

---

> ### Author Response · Authors · 2025-11-19
> **Regarding Question 1**
>
> ### **Regarding Question 1: How to guarantee that the reconstructed x' lies on the true data manifold?**
>
> This is a very deep and crucial theoretical question. We thank you for pushing us to clarify this fundamental point. First, to be precise, our method does not provide a **strict mathematical guarantee** that the reconstructed `x'` lies perfectly on the unknown true data manifold. In machine learning, the true manifold is typically inaccessible. However, the core advantage of FedMC is that it provides a **principled on-manifold approximation**, which is fundamentally different from existing approaches. The detailed explanation is as follows:
>
> 1.  **Calibration Occurs in the Tangent Space:** The principal component space learned by Kernel PCA is the best possible *local linear approximation* of the manifold at a data point, i.e., the **tangent space**.
> 2.  **Correctness of the Calibration Direction:** Our perturbation step occurs entirely *within* this tangent space. This means the direction of the generated augmented sample unfolds along the manifold's own geometric structure, rather than taking Euclidean "shortcuts" as traditional methods do.
> 3.  **Reconstruction with an "On-Manifold" Target:** The optimization objective of our pre-image reconstruction process is precisely that point in the high-dimensional feature space that lies "on the tangent space." Although solving the inverse problem involves an approximation, its target is explicitly defined to be "on-manifold."
>
> **The Fundamental Difference from Existing Methods:** Methods relying on the "global linearity assumption" have calibration directions that almost certainly deviate from the manifold, **systematically** generating out-of-distribution (OOD) samples. In contrast, every step in FedMC's design is intended to respect and adhere as closely as possible to the learned manifold structure. The significant performance gains we achieve in our experiments also provide strong empirical evidence for the effectiveness of this "on-manifold approximation" strategy.

---

> ### Author Response · Authors · 2025-11-19
> **Regarding Question 2**
>
> ### **Regarding Question 2: Why use Euclidean distance in the dynamic query?**
>
> You astutely point out a seeming contradiction in our use of Euclidean distance in Eq. 9 after critiquing it in Section 3. This is, in fact, a deliberate, **pragmatic, and reasonable design choice**. Euclidean distance plays two completely different roles in our framework, and its applicability differs accordingly:
>
> 1.  **Different Roles: Defining Geometry vs. Query Localization**
>     *   In **Section 3**, we criticize the use of Euclidean distance for **defining the intrinsic geometry of the manifold** (e.g., finding principal directions via global PCA). When the manifold is curved, a straight Euclidean line can cut through empty space where no data exists, which is incorrect.
>     *   However, in **Section 4.4**, the purpose of using Euclidean distance is much simpler and more direct: it serves as an efficient **"lookup" or "localization" tool**. Its task is to quickly find the local region to which the current data point `x` belongs within the global "geometry atlas" (the dictionary), represented by the nearest macro-prototype `g_l`.
>
> 2.  **Reasonableness of the Assumption:**
>     This "localization" task is based on a weaker and more reasonable assumption: **on a local scale, Euclidean distance is a good proxy for geodesic distance.** We are essentially using Euclidean distance to determine which "neighborhood" `x` shares with a prototype `g_l`. Once we have efficiently located the correct neighborhood, we immediately **switch to the complex, non-linear geometric model (learned by KPCA) of that neighborhood to perform the truly precise calibration.**
>
> In short, we use an efficient Euclidean distance for **"looking up the map"** (quickly selecting the correct local non-linear geometric template) and then use the complex geometric information on that map for precise **"navigation"** (performing on-manifold calibration). To compute the true geodesic distance for the "map lookup" step would not only be computationally prohibitive but would also lead to a circular problem: one needs to know the manifold to compute the distance, but the purpose of computing the distance is to learn the manifold. Therefore, our design strikes an ideal balance between computational efficiency and geometric fidelity.
>
> ---
>
> Once again, we thank you for your valuable time and professional advice. Based on your feedback, we will revise the manuscript to include the promised diagram and a detailed computational cost analysis, and to further clarify the theoretical underpinnings and design considerations of our method. We believe that with these revisions, the quality of the paper will be significantly enhanced, and that our responses fully address your concerns.

---

> > ### Comment · Reviewer_TzgE · 2025-11-27
> >
> > I have carefully reviewed the authors’ response, and I think most of my questions have been addressed. If the authors update the manuscript to incorporate these clarifications for Questions 1 and 2, I am willing to increase my score.

---

> > > ### Author Response · Authors · 2025-11-27
> > > **Thank you for your valuable contribution**
> > >
> > > **Dear Reviewer TzgE,**
> > >
> > > Thank you so much for your thoughtful and insightful comments—they have greatly helped us strengthen our paper. In direct response to your suggestions, we have incorporated the following key revisions into the manuscript (all newly added text is highlighted in blue in the revised submission):
> > >
> > > 1. **Clarification on the use of Euclidean distance in the Dynamic Geometry Query (after Eq. 9).**
> > >    We now explicitly state that the Euclidean distance here is **not used to define the intrinsic shape of the manifold**, but solely as a **computationally efficient and reasonable local proxy** for geodesic distance. Its purpose is to **quickly locate** the relevant geometric template for a given data point \(x\) within the global “geometry dictionary.” Once the appropriate neighborhood is identified, the actual manifold-aware calibration is performed using the **nonlinear structure** of the selected template—not the distance metric itself.
> > >
> > > 2. **Clarification on the on-manifold nature of the reconstructed sample \(x'\) (after Eq. 14).**
> > >    We added a concise theoretical clarification: although the true data manifold \(\mathcal{M}\) is unknown, our calibration is explicitly designed to produce an **on-manifold approximation**. The perturbation occurs in the **local tangent space** approximated by kernel PCA, ensuring the update direction aligns with the manifold’s intrinsic geometry rather than a global Euclidean shortcut. The pre-image reconstruction then seeks a point whose feature embedding matches this on-manifold target, thereby **avoiding the generation of out-of-distribution (OOD) samples** that plague global linear methods.
> > >
> > > 3. **Scalability and computational overhead analysis.**
> > >    As you suggested, we have added a new subsection (**Appendix B.3: Scalability and Computational Overhead Analysis**) that includes:
> > >    - A discussion of how **clustering bounds the input size** for local kernel PCA, controlling its \(\mathcal{O}(n^3)\) complexity;
> > >    - Empirical timing results showing that FedMC adds only **2.8s–4.1s per communication round** on large domain-skew datasets;
> > >    - Experiments with **100, 300, and 500 clients** on CIFAR-100 (\(\beta=0.1\)), demonstrating consistent performance gains and robustness at scale;
> > >    - An analysis of the favorable **compute–communication trade-off**: modest local computation enables faster convergence and higher accuracy, which is highly valuable in bandwidth-limited FL settings.
> > >
> > > 4. **Commitment to include a schematic illustration.**
> > >    We confirm that a **comprehensive figure** will be included in the camera-ready version to visually convey the core design of FedMC. The illustration will feature:
> > >    - An **S-shaped curved manifold** in the background to emphasize the nonlinear nature of real-world data;
> > >    - A clear **client–server split** showing the federated workflow across rounds;
> > >    - A dedicated icon for the **Global Anonymous Basis (GAB)**, with dashed arrows indicating its use in both geometry aggregation (Phase I) and calibration (Phase II);
> > >    - Labeled arrows showing the flow of key information (e.g., LGDs, geometry dictionary).
> > >
> > > We sincerely hope these revisions fully address your concerns and enhance the clarity, rigor, and reproducibility of our work. Thank you again for your time, expertise, and constructive feedback!

---

> > > > ### Comment · Reviewer_TzgE · 2025-11-27
> > > >
> > > > Thank you for the revision. I have raised my rating.

---

### Meta-Review · Area_Chair_aFCk · 2025-12-28

**Summary:**

This paper proposes Federated Manifold Calibration (FedMC), a novel framework that learns and leverages the local, nonlinear geometry of data.

All reviewers are inclined to accepting this paper. Authors should include the revisions into the new version properly, for example, revisions on method clarity, compuational analysis etc. Some specific comments include: It would be good to incorporate one diagram to help illustrate the key steps. ustify the scalability of FedMC in the settings where both the number of local data points and the number of clients may be large. Comment on the runtime and communication overhead of FedMC compared to standard FL baselines.

**Reviewer Concerns:**

Reviewers engaged in the rebuttal.

**Reviewer Scores:**

NA

---

### Decision · Program_Chairs · 2026-01-26

Accept (Poster)